# The Awareness of Pulmonologists and Patients with Respiratory Diseases about the Impact of Air Pollution on Health in Poland

**DOI:** 10.3390/jcm10122606

**Published:** 2021-06-12

**Authors:** Tadeusz M. Zielonka

**Affiliations:** Department of Family Medicine, Medical University of Warsaw, Lindleya Street 4, 02-005 Warsaw, Poland; tadeusz.zielonka@wum.edu.pl

**Keywords:** air pollution, awareness, education, knowledge, patients, pulmonologists, respiratory diseases

## Abstract

Within the European Union, air pollution is highest in Poland. The aim of this study was to compare the awareness of Polish pulmonologists and that of patients with respiratory diseases about the impact of air pollution on health. It was a crossover study with voluntary and anonymous participation. The study included 309 pulmonologists and 262 patients with respiratory diseases. The majority of the patients declared good knowledge about the impact of air pollution on health, and only 16% of the pulmonologists declared sufficient knowledge on this topic. The main sources of information on air pollution were radio and television for patients and the medical press for doctors. Doctors rarely informed patients about the impact of air pollution on their disease. Patients followed information on the quality of air in their areas more often than doctors. Polish patients’ knowledge about the main sources of air pollution in their areas was higher than the knowledge of pulmonologists. Patients declared knowledge of air pollution standards twice as often as doctors. Patients with respiratory diseases are interested in the effects of air pollution on their health. Polish patients’ knowledge about air pollution and its health effects is higher than that of the specialists treating them. Professional education of Polish pulmonologists in this field is needed.

## 1. Introduction

The latest report of the European Environment Agency indicates the highest concentrations of PM_10_ and PM_2.5_ in the European Union are found in Poland [1]. Every year in Poland, more than 48,000 people die prematurely due to air pollution, which is the highest rate in Europe in relation to a country’s population [2]. The main source of PM_10_ and PM_2.5_ in Poland is the household (over 50%) [3]. Air pollution results from the use of coal, often of very poor quality, for heating homes and flats. Therefore, social awareness of the health effects of air pollution is very important. The respiratory system is particularly exposed to the inhalation of polluted air [4,5]. Intense air pollution exacerbates lung diseases (e.g., asthma and chronic obstructive pulmonary disease—COPD), increasing the number of hospitalizations and deaths, with significant direct and indirect costs [6,7]. Air pollution is responsible for the development of obstructive pulmonary diseases [8,9]. The presence of pollutants in the home environment causes an increased incidence of respiratory infections [10]. The International Agency for Research on Cancer has classified atmospheric air pollution, especially, particulate matter, into the first group of human carcinogens [11]. This is a special challenge for pulmonologists. They should inform patients with chronic respiratory diseases about the harmful effects of air pollution on their health. They should also inform their patients about the need to apply appropriate measures to protect against over-exposure to air pollution [12]. Patients with respiratory diseases should be aware of the impact of air pollution on their health. These people especially should not contribute to increasing air pollution in their areas by burning rubbish in stoves, especially plastic packaging, which is very common in Polish homes.

The aim of the study was to compare the awareness of pulmonologists and patients with respiratory diseases about the impact of air pollution on health, as well as assess the sources of this knowledge and its implementations in practice.

## 2. Materials and Methods

A cross-sectional survey was conducted in Poland. Participation in the study was fully voluntary and anonymous. The study protocol was approved by the local ethics committee. Special questionnaires addressed knowledge about air pollution and also defined participants’ demographic and medical status (See Appendix A). There were small differences between the questionnaire version for physicians and that for patients. Patients were asked more questions. Patients were additionally asked to assess the actions of the government and local authorities in reducing pollution, the credibility of media information on air pollution, the personal protective equipment they use, as well as about the impact of air pollution on their health and whether they themselves can contribute to the improvement of air quality. For some questions, doctors responded by choosing yes or no, whereas patients indicated an option from 0 to 5 (e.g., 0—does not affect at all; 5—is extremely strong). When asked about the sources of knowledge, doctors and patients had different options to choose from. Finally, there were 3 identical questions verifying the knowledge of physicians and patients.

Patients were recruited from a private pulmonary disease outpatient clinic in Warsaw. They were people suffering from chronic lung diseases who agreed to participate in the study, completing the questionnaire while waiting for an appointment with a pulmonologist. During the visit, they had the opportunity to talk with the specialist on the topics covered in the survey. The study was carried out from 1 October 2018 to 30 June 2019. Pulmonologists participating in the study were recruited from doctors participating in training organized periodically by the author from 1 October 2017 to 30 November 2018. They accounted for about 20% of all pulmonologists in Poland. The questionnaires were distributed to participants of medical conferences, and their return was nearly 90%.

Information on the actual air pollution in the regions of the surveyed people was taken from the official website of the Chief Environmental Inspectorate in Poland [13].

A two-proportion z-test was used to test for a difference between two population proportions, with the null hypothesis (H0):H0: π1 = π2, assuming equal proportions in both independent groups. Assumptions concerning sample size for the z test for the difference between two proportions were checked. We chose z-test, and chi-square test for equality of two proportions was also considered; however, with one degree of freedom was considered exactly the same as a z-test. A value *p* < 0.05 was considered statistically significant.

## 3. Results

The study involved 262 patients (163 females and 99 males) with respiratory diseases, aged 61 ± 15 years (23–95 years). Seventy-six percent of them lived in Warsaw, and the others lived in the suburbs. The largest groups consisted of patients with asthma (56%) and COPD (27%), 10% of the patients had respiratory tract infections, 3% interstitial lung diseases, 2% lung cancer, and 2% with other respiratory diseases. Sixty-four percent of the patients reported comorbidities, mainly of the cardio-circulatory system. In the study, 309 pulmonologists also participated (216 females and 93 males) aged 51 ± 9.1 years (27–75 years). Pulmonologists came from various cities. The inhabitants of Warsaw constituted the largest group (17.5%).

Forty-six percent of the patients said that air pollution affected their health to a large or very large extent (Figure 1). Only 16% did not observe any effect of air pollution on their health. Significantly more patients (60%) reported a significant impact of air pollution on their health than negated it (Figure 1).

As many as 83% of the patients with respiratory diseases declared good, very good, or even excellent knowledge about the impact of air pollution on health (Figure 2), while only 16% of the pulmonologists declared sufficient knowledge about this topic.

The sources of this knowledge among physicians and patients appeared to be very different (Figure 3 and Figure 4). For patients, the media resulted to play a far more important role than information from medical personnel. Radio and television were most often indicated by patients as a source of information about the dangers of air pollution. Patients generally trusted the media. The vast majority (71.2%) of respondents did not think that press releases about the harmfulness of pollution to health were exaggerated, and only 8.5% were of the opposite opinion. Patients rarely indicated physicians, and especially general practitioners, as a source of information on the impact of pollution on health. In turn, for doctors, the most important source of knowledge was the medical press and the internet. Medical conferences were the preferred form of education in this topic by physicians (79.3%).

Patients declared knowledge of WHO and local air quality standards twice as often (*p* < 0.001) as doctors (Figure 5).

Patients’ declared and actual knowledge about the main sources of air pollution in their areas was much higher (*p* < 0.001) than the knowledge of pulmonologists (Figure 6).

Only 27.5% of the pulmonologists declared that they were following information on the level of air pollution in their area, and 16.5% of them were informing their patients about the consequences of air pollution on their health. Meanwhile, as many as 65% of patients systematically followed information on the quality of air in their regions and carried out their activities depending on them. Although the most common source of information about the current level of air pollution was TV, one-fourth of the patients who kept abreast of the concentration of pollution used an application on their mobile phones (Figure 7). As many as 69% of patients declared knowledge regarding methods of protection against air pollution. Thirty percent of the surveyed patients had an air purifier with a HEPA filter at home, and 5% of the patients used protective masks when leaving the house during periods of smog.

Physicians declared knowledge of diseases caused by air pollution more often than patients, but in reality, their knowledge on this subject was not greater (Figure 8). The difference between declared knowledge and actual knowledge of diseases caused by inhalation of PM_2.5_ for patients was only 2%, while for doctors, it was as much as 48%.

Patients knew the number of deaths caused by exposure to air pollution in Poland significantly more often than doctors (Figure 9). In general, patients correctly answered all three questions regarding knowledge more often than doctors and, at the same time, patients less often did not know the answer to any of the questions (Figure 9). However, the knowledge of patients was also very low, as only 8% of them correctly answered all three questions, and as many as 20% of them did not know the answers to any of the three questions. The differences in correct answers given by doctors and patients were statistically significant.

Only 4% of the patients evaluated the actions of local authorities as sufficient, and 7% of them positively assessed the actions of the government on this topic. As many as 42% of the patients thought they could also help reduce air pollution.

## 4. Discussion

The results only concern Poland and cannot be transferred to other countries. However, attention should be paid to the special position of Poland as regards air pollution in relation to other countries. Concentrations of benzo(a)pyrene as high as those in Poland are found in no other European country, and the same is true for concentrations of particulate matter, which are not as high as in Poland in the other European Union countries [1]. Poland is the only EU country that has not joined the European Green Deal proposed by the European Commission, the aim of which is to quickly reduce greenhouse gas emissions. More than 70% of the Polish energy sector is based on coal, and the share of renewable energy sources is the lowest in the European Union. Low public awareness of health risks related to climate change as well as air pollution certainly contributes to these bad air quality indicators in Poland. This presents a major challenge for physicians and other medical professionals, as they need to inform the public about the health risks associated with environmental changes. The presented results show that Polish doctors do not have adequate knowledge to cope with this task.

The obtained results shed new light on the problem of the causes of high air pollution in Poland. An important role for physicians is to inform patients about environmental health risks. Who, if not doctors and, especially, pulmonologists, should inform the public and local or central authorities about the impact of air pollution on health? The conducted research shows that Polish patients do not receive adequate help in this regard from their physicians. If Polish pulmonologists have less knowledge about the harmful effects of air pollution than their patients, if they do not know the standards for air pollution levels, and they do not follow the information on the concentration of air pollution in their areas, it is more difficult to improve the air quality in Poland.

There is little information about the knowledge of physicians regarding the impact of the environment on health. A French study, more than 20 years ago, showed a similar level of awareness of air pollution among French pulmonologists as that of Polish pulmonologists today [14]. It should be emphasized that it was then that the studies that showed a significant impact of air pollution on health were just beginning. Subsequently, studies in the US have shown that American physicians were much more aware of the health consequences of climate change and air pollution [15,16]. The knowledge of pulmonologists about the impact of environmental factors on the course of respiratory diseases is the basis for appropriate patient education, which for chronic diseases such as asthma or COPD, promotes beneficial health effects [17,18]. Monitoring the concentration of pollen in the air and informing the public allows individuals with atopy to avoid excessive exposure to allergens that cause asthma exacerbations. In the same way, patients with lung diseases should be informed about increases in air pollution, their health consequences, and how to protect against them [12,19].

This study has shown an extremely low level of knowledge and awareness of the air pollution problem by Polish pulmonologists. It is not physicians but non-governmental organizations such as the Smog Alarm, which inform Poles about the health-threatening situation regarding air pollution. In the Polish media, environmentalists and environmental engineers pay more attention to the health effects of air pollution than doctors. The obtained results suggest that this may be related to the lack of appropriate education of doctors. In the world over the last 20 years, knowledge about the impact of air pollution on health has developed very dynamically. The average age of Polish physicians, including specialists participating in the study, is quite advanced (over 50 years). Older doctors did not have the opportunity to learn about this problem in college. Perhaps, Polish medical universities have not modernized their study programs adequately to the rapid progress of science, and in postgraduate education, this topic is too rarely presented. This is indicated by the yet unpublished results of a research conducted by the author.

Greater awareness of Polish patients may result from the very high activity of the media, which have been devoting a lot of space to air pollution for several years now. Thanks to the radio, television, and the Internet, knowledge and public awareness in this area are growing. Many websites, TV, and radio stations in Poland provide constant information about the current state of air pollution. The pulmonologists participating in the study used these sources of information less frequently than the patients. Certainly, the media do not reach all environments with information about the harmfulness of air pollution, as evidenced by the low awareness of Polish inhabitants of villages and small towns, who, despite significant financial help, have not decided to replace old stoves with new, more ecological ones. Recently published research results in Malaysia showed a growing public awareness of the effects of air pollution, but a limited propensity to bear the costs associated with reducing exposure to these pollutants [20]. Only rich people or people with respiratory diseases in the family are willing to bear the costs for air pollution prevention control [20].

The vast majority of patients lived in Warsaw, the capital of the country with a population of 2 million, where air pollution is moderate. The city is dominated by pollution caused by car traffic. Thanks to modern central heating, pollution due to suspended dust (PM_2.5_ and PM_10_) is much lower than in the south of Poland. In other parts of the country, air pollution is much greater due to the heating of individual apartments with poor-quality coal, fired in old stoves. Therefore, the local level of air pollution is not the only factor that explains the high level of awareness of the examined patients. Polish patients with respiratory diseases are more interested in the problem of air pollution than their doctors. More often than their pulmonologists, they follow information on the pollution levels and try to protect against pollution. This indicates a great awareness of Polish patients who are trying to get to know the problem, because they feel its effects. These results are radically different from studies in India, where a lack of knowledge and low awareness of air pollution health risks among patients with asthma has been demonstrated [21]. Moreover, subsequent studies in China did not indicate a high level of awareness among Chinese citizens about the harmfulness of air pollution, despite the bad air quality in that country [22]. However, recently, a high knowledge awareness rate, strong health protection consciousness, and high enthusiasm for air pollution control among even the Chinese have been observed [23]. In a Chinese study, 48.5% of the participants used to wear face masks when going outside [23]. The study was conducted in Poland before the COVID-19 pandemic; therefore, the percentage of patients wearing protective masks was small. Unlike in Asian countries, very few people in Europe at that time used protective masks with HEPA filters against exposure to air pollution. It is surprising that in Poland as much as 30% of patients declared knowledge the WHO and national air quality standards. However, it was only a declaration of knowledge and not an exact knowledge of the numerical values for each type of pollutant.

The limitation of the study regards the recruitment of the participants. Concerning the pulmonologists, they were attending a medical training course, which led to a certain selection of doctors who learn and expand their knowledge. It cannot be ruled out that the results for doctors not involved in continued learning would be worse. Patients, on the other hand, were only from one private outpatient’s clinic in Warsaw. They were, therefore, health-conscious people who could afford the use of the private medical sector. It can be assumed that in the general population, including both doctors and patients, the awareness of the risks associated with exposure to air pollution is lower. Relatively high interest of Polish patients in the problem of air pollution may result from their respiratory system diseases. Moreover, they were well-off and health-conscious people. In Malaysia, patients who had respiratory disease or had been hospitalized showed no significant difference in their overall awareness level compared to those without respiratory diseases [20]. Pulmonologists completed the questionnaires two years earlier than the patients. As the problem of air pollution has become very popular in various media in Poland in recent years, it cannot be ruled out that the knowledge of physicians on this subject has improved significantly. It is worth repeating the study to check if this is true. Another limitation of this study was basing the survey on respondents’ declarations. The verification questions showed that especially doctors declared more knowledge than they actually possessed much more often than patients. It can therefore be assumed that their actual knowledge is much worse. This is a single-country evaluation of patient and physician self-reported knowledge regarding air pollution and its impact on health. It is certainly worth carrying out a similar assessment among physicians and patients in other countries.

## 5. Conclusions

Polish patients with respiratory diseases are interested in the effects of air pollution on their health. Polish pulmonologists know little about air pollution. Professional training of Polish pulmonologists in this field is needed.

## Figures and Tables

**Figure 1 jcm-10-02606-f001:**
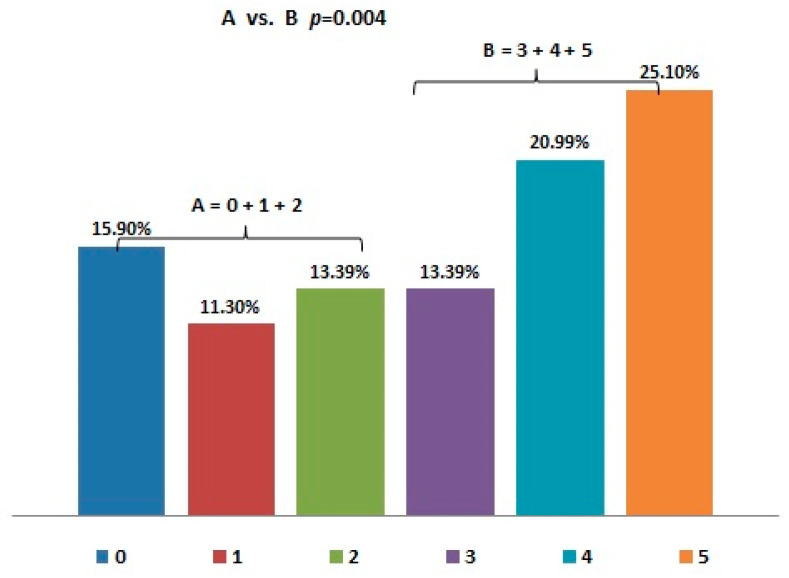
Assessment of the impact of air pollution on their health by patients with lung diseases (from 0—no influence of pollution to 5—very strong influence). The study group was divided into two subgroups. Group A consisted of patients declaring no impact, very weak, and weak impact of air pollution on their health, while group B consisted of patients who declared moderate, high, and very high impact.

**Figure 2 jcm-10-02606-f002:**
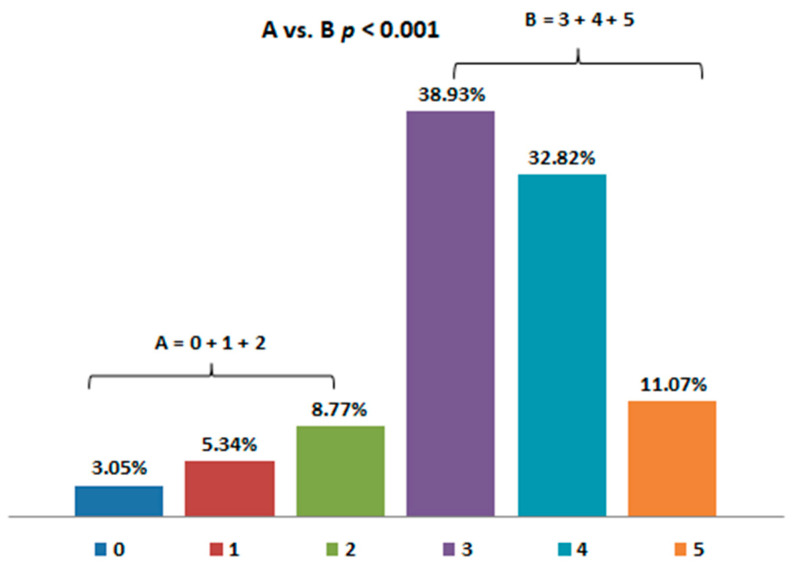
Patients’ self-assessment of knowledge about the impact of air pollution on health (from 0—lack of knowledge to 5—very extensive knowledge). The study group was divided into two subgroups. Group A consisted of patients declaring lack of knowledge, very poor, and poor knowledge about the impact of pollution on health, while group B consisted of patients who declared good, very good, or excellent knowledge on this subject.

**Figure 3 jcm-10-02606-f003:**
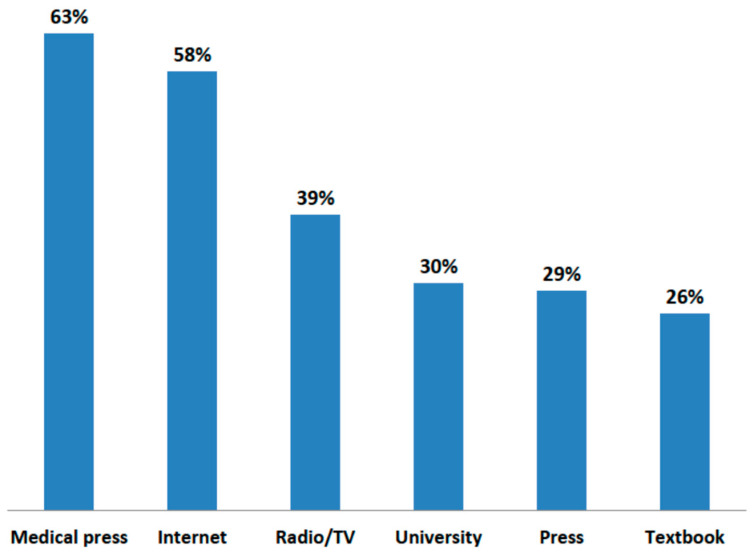
Main sources of pulmonologists’ knowledge about the impact of air pollution on health.

**Figure 4 jcm-10-02606-f004:**
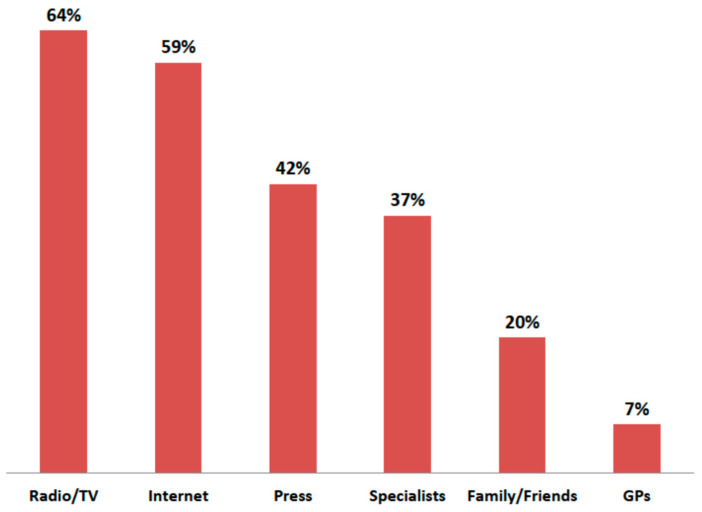
Main sources of patients’ knowledge about the impact of air pollution on health.

**Figure 5 jcm-10-02606-f005:**
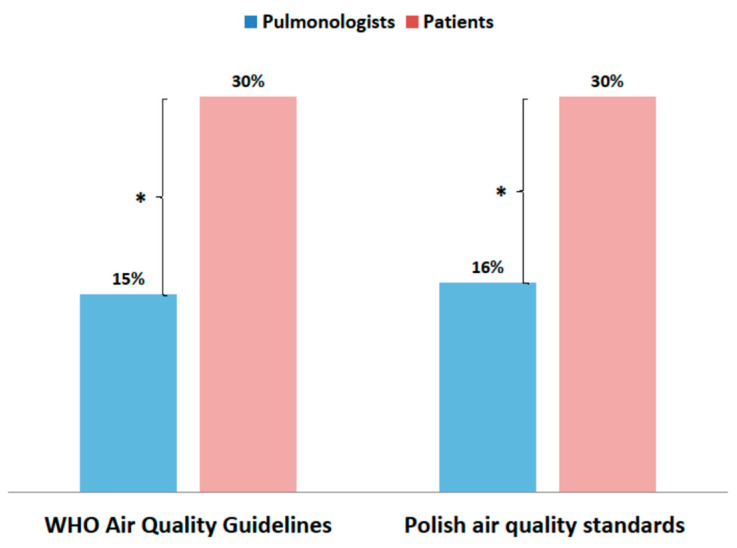
Declared knowledge of WHO and Polish air quality standards by patients and pulmonologists (* *p* < 0.001).

**Figure 6 jcm-10-02606-f006:**
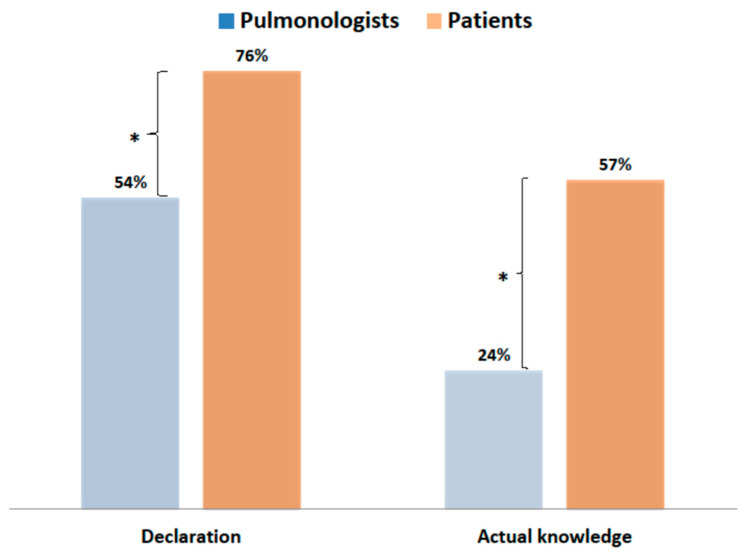
Comparison of the knowledge of patients and that of physicians about the main sources of air pollution in their areas (* *p* < 0.001).

**Figure 7 jcm-10-02606-f007:**
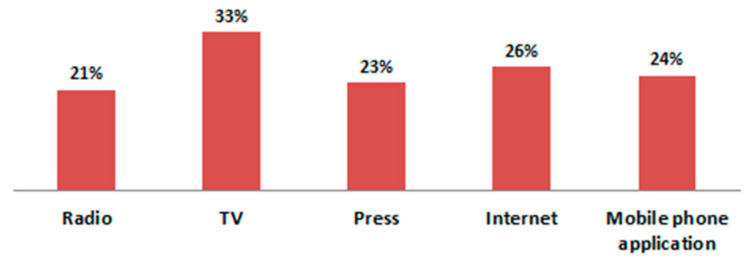
Sources of information used by patients regarding the level of air pollutants in their areas.

**Figure 8 jcm-10-02606-f008:**
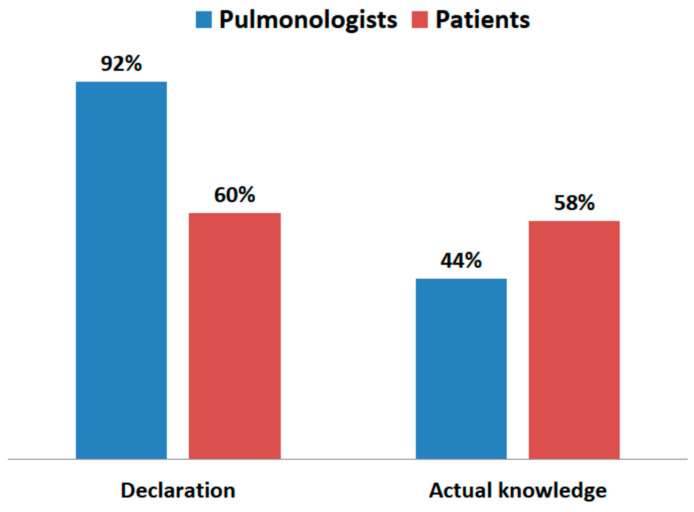
Declared and actual knowledge of patients and physicians about diseases caused by air pollution.

**Figure 9 jcm-10-02606-f009:**
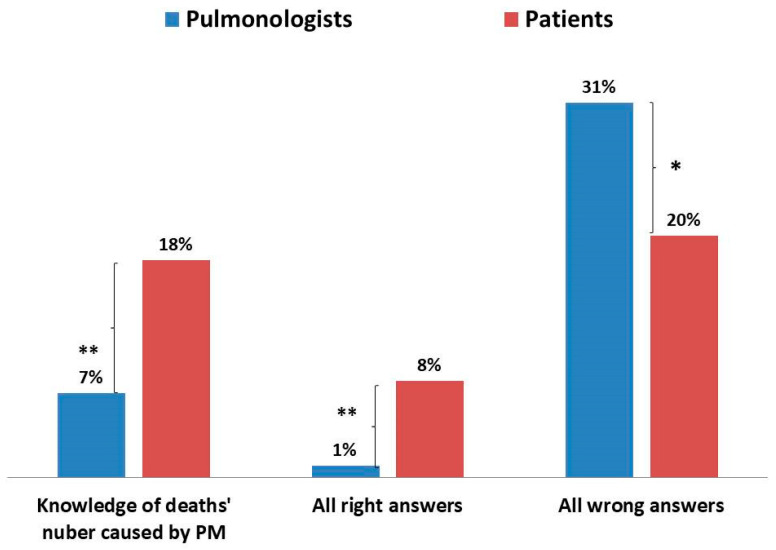
Patients’ and pulmonologists’ global knowledge about the impact of air pollution (*, ** *p* < 0.001).

## Data Availability

Data is contained within the article or Appendix A.

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
