# Peer review of "The Awareness of Pulmonologists and Patients with Respiratory Diseases about the Impact of Air Pollution on Health in Poland"

_jcm, 2021, doi:10.3390/jcm10122606_

Round 1

Reviewer 1 Report

Overall comments

It would be useful to add the 2 types of questionnaires as Supplementary Material.

As the study was conducted in Warsaw, it may be useful to talk briefly about the local air quality as it may explain (or not) the relatively high knowledge amongst patients.

I think it would very useful to explain how ‘actual’ or ‘real’ knowledge was determined. This could be done by giving some example questions, or of course, if the questionnaires were included as Supplementary Material.

The Discussion is a bit repetitive. It makes a point and then revisits the same point again at a later date. For example, after saying ‘It is related to the lack of proper education of physicians’ on line 149, it would be good to immediately follow this up with the information on lines 166-168 rather than re-introducing the subject again.

Much of the discussion is devoted to the lack of knowledge of pulmonologists but not much is said about what appears to be a high degree of knowledge amongst patients – for example, what motivates them to keep up to date with air pollution data?

In the final paragraph, some discussion would be interesting on reasons why awareness in Poland is greater than India and also to include what the situation is like in countries with better air quality such as the USA.

Specific comments

Line 33: Explain what is meant by ‘low emission’.

Line 36: Define ‘smog’.

Line 36: Define COPD abbreviation upon first use.

Line 44: The use of the word ‘monitor’ is questioned. This implies that patients are actually measuring air pollutant concentrations. I think it should be substituted with ‘be aware of the level of air pollution’.

Figures 1 & 2: Suggest that the individual 0-5 rankings are spelt out in a key rather than just defining ‘0’ and ‘5’ in the title.

Lines 98-101: Refer to WHO Air Quality Guidelines rather than ‘norms’ or standards. And refer to Polish air quality standards rather than ‘norms’.

Line 114: Again ‘monitor’ is probably used incorrectly unless the HEPA also give a read out of pollutant concentrations. Suggest substituting with ‘keep abreast of’

Line 125: ‘Decelerated’ should read ‘Declared’.

Line 127-130: Even though a greater percentage of patients answered the 3 questions correctly, it was still low (8%). I think this should be said.

Line 140: Replace ‘standards’ with ‘guidelines and standards’.

Line 142: Reword/clarify what is meant by ‘difficult to count on a good situation in the field of air pollution’.

Line 149: The statement ‘It is related to the lack of proper education of physicians’ is a very black and white one. Can it be referenced?

Line 163: Replace ‘di’ with ‘do’.

Line 164-165: Qualify where this increased knowledge is – ie within general population.

Lines 173-176: This sentence needs attention. It reads as though air pollution has beneficial effects.

Line 176: Suggest ‘monitoring’ is replaced with ‘awareness of’.

Line 199: Give some examples of these costs.

Line 202: Ref 22 is not in the list.

Author Response

Overall comments

It would be useful to add the 2 types of questionnaires as Supplementary Material.

Of course, the questionnaire for physicians and second for patients will be added as Supplementary Data.

As the study was conducted in Warsaw, it may be useful to talk briefly about the local air quality as it may explain (or not) the relatively high knowledge amongst patients.

The vast majority of respondents live in Warsaw, the capital of the country with a multimillion population, where air pollution is moderate. The city is dominated by pollution caused by car traffic. Thanks to the modern heating center, the pollution with suspended dust (PM2.5 and PM10) is much lower than in South of Poland. In other parts of the country, air pollution is much greater due to the heating of individual apartments with poor-quality coal fired in old stoves. Therefore, the local level of air pollution does not explain the high level of examined patients’ awareness.

I think it would very useful to explain how ‘actual’ or ‘real’ knowledge was determined. This could be done by giving some example questions, or of course, if the questionnaires were included as Supplementary Material.

So that there are no doubts about the real and declared knowledge, questionnaires with all questions will be added as Supplementary Material.

The Discussion is a bit repetitive. It makes a point and then revisits the same point again at a later date. For example, after saying ‘It is related to the lack of proper education of physicians’ on line 149, it would be good to immediately follow this up with the information on lines 166-168 rather than re-introducing the subject again.

I agree with this opinion and have changed the location of the indicated sentences.

Much of the discussion is devoted to the lack of knowledge of pulmonologists but not much is said about what appears to be a high degree of knowledge amongst patients – for example, what motivates them to keep up to date with air pollution data?

Relatively high interest of Polish patients in the problem of air pollution may result from their respiratory system diseases. Moreover, they were well-off and health-conscious people. In Malaysia, patients who had respiratory disease or had been hospitalized showed no significant difference in overall awareness level compared to those who did not [22].

In the final paragraph, some discussion would be interesting on reasons why awareness in Poland is greater than India and also to include what the situation is like in countries with better air quality such as the USA.

I supplemented the discussion with the indicated threads.

I supplemented the discussion with the indicated threads.: Greater awareness of Polish patients may result from the very high activity of the media, which have been devoting a lot of space to air pollution for several years now. Many websites, TV and radio stations provide constant information about the current state of air pollution. Unfortunately, busy doctors do not have the opportunity to read the information commonly available to their patients.

Certainly, the media do not reach all environments with information about the harmfulness of air pollution, as evidenced by the low awareness of Polish inhabitants of villages and small towns, who, despite a significant financial help, do not decide to replace old stoves with new, more ecological ones.

Specific comments

Line 33: Explain what is meant by ‘low emission’.

I used the Polish term “low emission”, i.e. up to 40 meters high. I replaced this term with a more understandable expression: “The main source of PM10 and PM2.5 in Poland is households sector (over 50%) [3].”

Line 36: Define ‘smog’

I presented in the introduction more precise that “Smog (smoke fog) as a type of intense air pollution exacerbates lung diseases ….”

The meaning of this abbreviation is defined in the text of paper.

Line 44: The use of the word ‘monitor’ is questioned. This implies that patients are actually measuring air pollutant concentrations. I think it should be substituted with ‘be aware of the level of air pollution’.

You are right. I changed it.

Figures 1 & 2: Suggest that the individual 0-5 rankings are spelt out in a key rather than just defining ‘0’ and ‘5’ in the title.

You are right. I changed it.

Lines 98-101: Refer to WHO Air Quality Guidelines rather than ‘norms’ or standards. And refer to Polish air quality standards rather than ‘norms’.

It is true. I changed it.

Line 114: Again ‘monitor’ is probably used incorrectly unless the HEPA also give a read out of pollutant concentrations. Suggest substituting with ‘keep abreast of’

I agree with you. I changed it.

Line 125: ‘Decelerated’ should read ‘Declared’.

I corrected a spelling mistake (thank you).

Line 127-130: Even though a greater percentage of patients answered the 3 questions correctly, it was still low (8%). I think this should be said.

I have corrected this sentence as you suggested

Line 140: Replace ‘standards’ with ‘guidelines and standards’.

You are right. I changed it.

Line 142: Reword/clarify what is meant by ‘difficult to count on a good situation in the field of air pollution’.

I have tried to present this problem more precisely. I hope that it is better understood after the changes.

Line 149: The statement ‘It is related to the lack of proper education of physicians’ is a very black and white one. Can it be referenced?

I corrected this sentence: “The obtained results suggest that it may be related to the lack of appropriate education of doctors.”

Line 163: Replace ‘di’ with ‘do’.

I corrected a spelling mistake (thank you).

Line 164-165: Qualify where this increased knowledge is – ie within general population.

I corrected it, it was about general knowledge on the subject.

Lines 173-176: This sentence needs attention. It reads as though air pollution has beneficial effects.

You were right. The term "patient education" was missing in this sentence. After the sentence was corrected, it took on its proper meaning: “The knowledge of pulmonologists about the impact of air pollution on the course of respiratory diseases is the basis of appropriate patients education, which in chronic diseases such as asthma or COPD causes beneficial health effects [16].”

Line 176: Suggest ‘monitoring’ is replaced with ‘awareness of’.

I have consistently changed this entry as above.

Line 199: Give some examples of these costs.

The authors of the article did not present what costs they are ready for. They only examined the willingness to pay the additional costs associated with air pollution control. However, this sentence was modified to accurately reflect the result of the authors of the article.

Line 202: Ref 22 is not in the list.

It is true, there was an incorrect presentation  of references. I lost item 12. After adding it, the numbering of all the next ones changed. Thank you very much for noticing this error.

Reviewer 2 Report

This is a simple survey of Polish respiratory patients and physicians, designed to explore their KAP in relation to air pollution.  There appears to be a surprising degree of ignorance among the doctors.  The manuscript is fairly clear but it is missing some important detail, and I would make the following suggestions:

  1. Without information on the denominators (which I imagine is difficult to get now) we cannot know how representative this is; I suggest mention of this is made in the Discussion
  2. It would greatly help the reader to have access to the questionnaire(s); can they be included in a supplement? Without this it is impossible to understand a lot of what follows, such as ‘declared’ and ‘actual’ or ‘real’ knowledge.
  3. Introduction, line 32; what does ‘the main source of … is low emission …’ mean?
  4. A statistical test on the data in Figures 1 and 2 is meaningless and I suggest the p values are dropped. I would use % (not ‘n’) at the top of each column, as you have done in the remaining Figures
  5. I’m impressed that 30% of patients know the WHO and national air quality standards; surely this can’t literally be the case? Which standards (PM, NOx etc)?  Again, a reason to publish the questionnaire.
  6. Presumably the 5% figure for patients using masks during periods of ‘smog’ was pre-Covid; it would help to include the date of the survey somewhere.
  7. In the Discussion there are several references to ‘professional medicine’; this is more normally referred to as ‘occupational medicine’ in English
  8. The final page of the MS (listing reference 22) is missing

Author Response

Thank you very much for all your comments and suggestions that are very helpful for improving the manuscript. I have corrected the text in accordance with all your remarks

Ad 1. Without information on the denominators (which I imagine is difficult to get now) we cannot know how representative this is; I suggest mention of this is made in the Discussion

It is true that the limitation of this work is the ignorance of the representativeness of the studied groups. In the case of doctors, they were attending medical training courses and they accounted for about 10% of all pulmonologists in Poland. Patients, on the other hand, were only from private outpatients clinics in Warsaw. They were therefore health-conscious people who could afford the use of the private medical sector. It can be assumed that in the general population, both doctors and patients, the awareness of the risks associated with exposure to air pollution is lower. This information was added to the discussion.

Ad 2. It would greatly help the reader to have access to the questionnaire(s); can they be included in a supplement? Without this it is impossible to understand a lot of what follows, such as ‘declared’ and ‘actual’ or ‘real’ knowledge.

The questionnaire for physicians and second for patients will be added as Supplementary Data.

Ad 3. Introduction, line 32; what does ‘the main source of … is low emission …’ mean?

In the introduction, I used the Polish term “low emission”, i.e. up to 40 meters high. I replaced this term with a more understandable expression: “The main source of PM10 and PM2.5 in Poland is households sector (over 50%) [3].”

Ad 4. A statistical test on the data in Figures 1 and 2 is meaningless and I suggest the p values are dropped. I would use % (not ‘n’) at the top of each column, as you have done in the remaining Figures

I changed figure 1 and figure 2 according to your suggestions:

Fig. 1 new

Ad 5. I’m impressed that 30% of patients know the WHO and national air quality standards; surely this can’t literally be the case? Which standards (PM, NOx etc)?  Again, a reason to publish the questionnaire.

It is surprising that as much as 30% of patients know the WHO and national air quality standards. However, it was only a declaration of knowledge and, presuming surely, it was not a precise knowledge of the numerical values for each pollutant type.

Ad 6. Presumably the 5% figure for patients using masks during periods of ‘smog’ was pre-Covid; it would help to include the date of the survey somewhere.

You are right, the study was conducted before the COVID-19 pandemic. I added information on this in the Methods section and I commented on it in the discussion.

Ad 7. In the Discussion there are several references to ‘professional medicine’; this is more normally referred to as ‘occupational medicine’ in English

I corrected the term “professional medicine” to “occupational medicine” in the discussion.

Ad 8. The final page of the MS (listing reference 22) is missing

It is true, there was an incorrect presentation  of references. I lost item 12. After adding it, the numbering of all the next ones changed. Thank you very much for noticing this error.

Reviewer 3 Report

This is a single country evaluation of patient and physician reported knowledge regarding air pollution and impact on health.

Not novel or unique or generalisable to a wider audience

The paper is of interest however there are serious flaws and significant editing required for this paper to be acceptable for publication. The paper is poorly written and presented and needs to be re-written. There is lack of detail on specific themes that were to be questioned in the questionnaires. I enclose specific comments below

Introduction:

Describes the impact of air pollution on health with specific references and sets the scene for the study.

Aims described

Methods:

This section needs significant improvement as jumps from methods to results

Single country survey in Poland.

Description about survey differences between physicians and patients not clear and could describe themes related to the questionnaire. How were questions derived? what specific questions were asked? Why asked?

Patients recruited from pulmonary clinics in Warsaw – what type of clinics? How many hospitals? Were all patient invited? Over what time period.

Physicians recruited from meetings? How? how many asked? how many replied? Is it representative?

Author presents results in Methods section – please remove demographics to results section.

Need to describe statistics clearer

Add the questionaires in a supplement

Results

Figure 1 not well labelled. All figures appear in text with limited explanations

How does air pollution affect their health?

Figures are just presented with minimal text.

Discussion:

Not generalisable; specific to Poland. 

Author Response

Thank you very much for all your critical comments and suggestions, which were very helpful in improving the manuscript, have allowed me to look more broadly and fully at the results obtained and allowed also to significantly improve the text of the manuscript by following these advices. I prepared also much bigger group of physicians. I presented much more information concerning methods.

Methods:

This section needs significant improvement as jumps from methods to results

I revised and supplemented methods section according to the reviewers' instructions.

Single country survey in Poland.

This is a single country evaluation of patient and physician reported knowledge regarding air pollution and impact on health. It is certainly worth carrying out a similar assessment among physicians and patients in other countries.

Description about survey differences between physicians and patients not clear and could describe themes related to the questionnaire. How were questions derived? what specific questions were asked? Why asked?

Special questionnaires addressed knowledge about air pollution, and also defined participants’ demographic and medical status (See Supplementary Materials). There were only small differences between the version for physicians and for patients. Patients were asked more questions. Patients were additionally asked to assess the actions of the government and local government in reducing pollution, to assess the credibility of media information on air pollution, the personal protective equipment they use, about the impact of air pollution on their health and whether they themselves can contribute to the improvement of air quality? In some questions, doctors had a choice of yes or no and patients indicated an option from 0 to 5 (0 - I don't know anything, 5 - I know a lot). When asking about the sources of knowledge, doctors and patients had different options to choose from. Finally, there were 3 identical questions verifying the knowledge of physicians and patients.

For the avoidance of doubt regarding differences in the questionnaire between physicians and patients, both questionnaires will be submitted in full as Supplementary Materials.

Patients recruited from pulmonary clinics in Warsaw – what type of clinics? How many hospitals? Were all patient invited? Over what time period.

Patients were recruited from one pulmonary diseases outpatient clinic in private sector in Warsaw. They were people suffering from chronic lung diseases who agreed to participate in the study, completing the questionnaire while waiting for an appointment with a pulmonologist. During the visit, they had the opportunity to talk with the specialist on the topics covered in the survey. It can be said that, in principle, they were the next patients who came to the clinic because there were very rare refusals, dictated by the lack of glasses, poor eyesight, etc. The study was carried out from 1 October 2018 to 30 June 2019.

Physicians recruited from meetings? How? how many asked? how many replied? Is it representative?

Pulmonologists participating in the study were recruited from doctors participating in the training organized periodically by the author from 1 October 2017 to 30 November 2018. They accounted for about 20% of all pulmonologists in Poland. The questionnaires were distributed to participants of medical conferences and their return was nearly 90%.

In addition, the limitations caused by the imperfect representativeness of the study participants were discussed in the discussion.

Author presents results in Methods section – please remove demographics to results section.

Patient and physician demographics have been removed from the Methods to the Results section.

Need to describe statistics clearer

A two proportion z-test was used to test for a difference between two population proportions, with following null hypothesis (H0):H0: π1=π2 assuming equal proportions in both independent groups. Assumptions concerning sample size for the z test for the difference between two proportions were checked. We chose z-test and chi-square test for equality of two proportions was also considered, however, with one degree of freedom is considered exactly the same as a z-test. A value p < 0.05 was considered statistically significant.

  1. Add the questionaires in a supplement

The questionnaire for physicians and second for patients will be added as Supplementary Data.

Results

  1. Figure 1 not well labelled. All figures appear in text with limited explanations

As you suggest, Figure 1 and Figure 2 have been precisely labeled. In addition, the figures in the text are explained in more detail.

Figure 1 Assessment of the impact of air pollution on their health by patients with lung diseases (0 - no influence of pollution; 5 - strongest influence). The study group was divided into 2 subgroups. Group A consisted of patients declaring no impact, very weak and weak impact of air pollution on their health. While, group B consisted of patients who declared moderate, high and very high impact.

Figure 2 Assessment of patients' knowledge about the impact of air pollution on health (0 - lack of knowledge; 5 - very extensive knowledge). The study group was divided into 2 subgroups. Group A consisted of patients declaring lack of knowledge, very poor and poor knowledge about the impact of pollution on health. While, group B consisted of patients who declared good, very good or excellent knowledge on this subject.

  1. How does air pollution affect their health?

Forty-seven percent of patients said that air pollution their health to a large or very affects large extent (Figure 1). They observed exacerbations of chronic respiratory diseases in the period of deteriorating air quality in their environment. Only 16% did not observe any air pollution effect on their health. Significantly more patients (60%) reported a significant impact of air pollution on their health than negated it.

  1. Figures are just presented with minimal text.

On the advice of the reviewer, the description of the obtained results was expanded

Discussion:

  1. Not generalisable; specific to Poland.

I eliminated generalizations from the discussion, emphasizing that the results only concern Poland and cannot be transferred to other countries. At the same time, I drew attention to the special position of Poland in the field of air pollution in relation to other European Union countries. In line with the reviewers' comments, the discussion was completed and modified.

The results only concern Poland and cannot be transferred to other countries. However, attention should be paid to the special position of Poland in the field of air pollution in relation to other European Union countries. Such high concentrations of bezno(a)pyrene are found in no European country and such high concentrations of particulate matter are not observed in any of the European Union countries [1]. Poland is the only EU country that has not joined the green deal proposed by the European Commission, the aim of which is to quickly reduce greenhouse gas emissions. More than 70% of the Polish energy sector is based on coal and the share of renewable energy sources is the lowest in the European Union. Low public awareness of health risks related to climate change as well as air pollution certainly contributes to these bad air quality indicators in Poland. This presents a major challenge for physicians and other medical professionals as they need to inform the public about the health risks associated with environmental changes. The presented results show that Polish doctors do not have adequate knowledge to cope with this task.

Round 2

Reviewer 3 Report

Thank you for revising the publication. You have revised the publication in line with reviewers comments. There are however some additional findings and comments that need to be addressed:

Please change the title to ….In Poland (at the end of the title)

Intro Line 59 This people not grammatically correct please revise

Materials and methods Line 84 Remove the sentence “It can be said that, in…”

Results:

Figure 2 is left justified and needs to be central on the page

Figure 3 and 4 would benefit from displaying the most frequent sources first on the graph to the least frequent as this would be easier to read

The text about figure 3 and 4 would be best to be before the figures

For figures 5  - you haven’t described how you defined actual air pollution in the methods – ie where are readings taken from what dates etc

Discussion

Without evidence or references, I think it is not accurate for the authors to say local Polish medical education doesn’t cover air pollution. This cannot be corroborated. This needs to be toned down as a reason. Polish doctors don’t have time to follow the news – again not evidence based and just an assumption – please revise your argument here as its all assumptions about Polish physicians education.

First line of limitations – I do not understand – please revise or remove. Again a lot of assumptions are made here. Please avoid making assumptions in the discussion about levels of knowledge of patients not studied here and just present the facts

Discussion needs revising to avoid assumptions about physicians and patients and just give the facts – or reference your assumptions.

The final paragraph of the discussion is very important and needs to be at the beginning of the discussion or even in the introduction as it justifies the rationale for this study

Author Response

Thanks again for all valuable remarks and comments

Please change the title to ….In Poland (at the end of the title)

Yes, it is indeed a necessary change.

Intro Line 59 This people not grammatically correct please revise

Of course I corrected this error.

Materials and methods Line 84 Remove the sentence “It can be said that, in…”

Yes. I did it.

Results:

Figure 2 is left justified and needs to be central on the page

Yes, I changed it.

Figure 3 and 4 would benefit from displaying the most frequent sources first on the graph to the least frequent as this would be easier to read

Yes. I did it. You are absolutely right. The new version is certainly better for the reader. In the original version, the order in the questionnaire was presented.

The text about figure 3 and 4 would be best to be before the figures

Yes. I did it.

For figures 5  - you haven’t described how you defined actual air pollution in the methods – ie where are readings taken from what dates etc

You are right. I made up this in the methods part.

Information on the actual air pollution in the regions of the surveyed people was taken from the official website of the Chief Environmental Inspectorate in Poland. (http//powietrze.gios.gov.pl/pjp//home)

Discussion

Without evidence or references, I think it is not accurate for the authors to say local Polish medical education doesn’t cover air pollution. This cannot be corroborated. This needs to be toned down as a reason. Polish doctors don’t have time to follow the news – again not evidence based and just an assumption – please revise your argument here as its all assumptions about Polish physicians education.

Thank you once again for the in-depth re-evaluation of the paper and for pointing out its weaknesses. Some matters are so obvious and well known in Poland that I forget that they are not obvious to people from other countries. For example, there are no doctors in Poland. Although doctors work until death without retiring, we have the lowest number of doctors per population in Europe. This results doctors' overwork and lack of time. The official average working time of doctors in Poland is 80 hours a week. Many doctors work more than 120 hours a week. While the working time permitted in the European Union may not exceed 48 hours a week. Polish doctors work in several places. I myself work as a doctor in five different places, often from 8 am to 10 pm non-stop. It's all about work, not readiness to work. The lack of time of Polish doctors is as obvious as the need to breathe. Doctors' overwork is a serious threat to their postgraduate education. You are also right in pointing out that the results discussed do not support the claim that the university education of doctors in the field of the impact of air pollution is insufficient. It was shown by my next, yet unpublished research, which I am currently conducting. However, knowing that this is the case, I have no reason to write about it based on the previous results presented in this paper. As recommended, I have modified the discussion by weakening overly far-reaching statements.

First line of limitations – I do not understand – please revise or remove. Again a lot of assumptions are made here. Please avoid making assumptions in the discussion about levels of knowledge of patients not studied here and just present the facts

I have modified this sentence.

The limitation of the study is the recruitment of participants

Discussion needs revising to avoid assumptions about physicians and patients and just give the facts – or reference your assumptions.

I checked the discussion once again, eliminating from it overly authoritative statements.

The final paragraph of the discussion is very important and needs to be at the beginning of the discussion or even in the introduction as it justifies the rationale for this study

I moved this paragraph at the beginning of the discussion, in accordance with your recommendations.
